# Rhinovirus and Cell Death

**DOI:** 10.3390/v13040629

**Published:** 2021-04-07

**Authors:** Shannic-Le Kerr, Cynthia Mathew, Reena Ghildyal

**Affiliations:** Faculty of Science and Technology, University of Canberra, Canberra 2617, Australia; shannic.kerr@canberra.edu.au (S.-L.K.); Cynthia.Mathew@canberra.edu.au (C.M.)

**Keywords:** rhinovirus, cell death pathways, lifecycle, apoptosis, necrosis, necroptosis, autophagy

## Abstract

Rhinoviruses (RVs) are the etiological agents of upper respiratory tract infections, particularly the common cold. Infections in the lower respiratory tract is shown to cause severe disease and exacerbations in asthma and COPD patients. Viruses being obligate parasites, hijack host cell pathways such as programmed cell death to suppress host antiviral responses and prolong viral replication and propagation. RVs are non-enveloped positive sense RNA viruses with a lifecycle fully contained within the cytoplasm. Despite decades of study, the details of how RVs exit the infected cell are still unclear. There are some diverse studies that suggest a possible role for programmed cell death. In this review, we aimed to consolidate current literature on the impact of RVs on cell death to inform future research on the topic. We searched peer reviewed English language literature in the past 21 years for studies on the interaction with and modulation of cell death pathways by RVs, placing it in the context of the broader knowledge of these interconnected pathways from other systems. Our review strongly suggests a role for necroptosis and/or autophagy in RV release, with the caveat that all the literature is based on RV-A and RV-B strains, with no studies to date examining the interaction of RV-C strains with cell death pathways.

## 1. Introduction

Programmed cell death is a key component of the host antiviral response, but picornaviruses, including rhinoviruses (RVs), are able to modulate cell death at different stages of virus lifecycle; inhibition of apoptosis early in infection facilitates virus survival, while induction of apoptosis later in infection may aid virus release. In addition to apoptosis, we now know that there are several kinds of programmed cell death, e.g., necroptosis, pyroptosis, ferroptosis and parthanatos; all of which are implicated in one or more virus infections [1,2,3,4]. With our increasing understanding of the complexities of cell death pathways, and the host–virus interactions during RV infection, it is becoming clear that RV components interact with, interrupt or modulate several cell-signaling cascades in the infected cell to support the virus lifecycle [1,2,3,4,5].

In this review we aim to summarize current literature on the interaction of RV with cell death pathways to enable replication and virus release in order to integrate diverse studies to inform future research.

## 2. Search Strategy

PubMed, Scopus and Google Scholar were extensively searched for research papers published between 2000 and 2021 using a MeSH database search strategy. The keywords used for the search were ‘rhinovirus’ and ‘apoptosis’ or ‘necrosis’ or ‘necroptosis’ or ‘cell death pathways’ or ‘cell death mechanism’. Furthermore, a small number of additional articles were identified from the reference lists. The search was limited to papers that were available in English and in full text.

## 3. Virion Structure and Lifecycle

RV is the leading cause of upper respiratory infections globally, affecting millions of people every year [6,7,8,9]. RV infections can be asymptomatic, cause mild upper respiratory tract symptoms or lead to severe lower respiratory tract infections and exacerbations in asthma and COPD [10,11]. RV is a positive sense RNA virus belonging to the *Enterovirus* genus in the *Picornaviridae* family [6,12]. RVs are highly diverse and to date, 170 strains have been described based variously on serotyping and/or genetic sequencing. RVs can be classified into three species (A, B, C) based on gene sequences. RVs belonging to A and B species can be further subclassified into major or minor groups based on their receptor usage [13,14].

### 3.1. Virion Structure

RVs are non-enveloped single-stranded(+) RNA viruses. The structural proteins VP1, VP2, VP3, and VP4 form an icosahedral capsid that encases the RNA genome [15]. The capsids are composed of 60 copies each of VP1-4. The VP1-3 confer strain-specific properties of immunogenicity, receptor binding and drug susceptibility to each RV isolate. The short VP4 proteins localize inside the capsid, attached to the genome. The single stranded RV genome (approximately 7200 bp) consists of a single open reading frame with a 5′ untranslated region and a short viral priming protein (VPg) that acts as a primer for replication [13,14,15]. 

### 3.2. Viral Replication, Assembly and Release

RV infection typically involves the following sequence (Figure 1). (1) Binding of the virus to the respective cognate receptors on the plasma membrane facilitated by VP1. Majority of known RV strains (most RV-A and all RV-B) target ICAM1, a type 1 transmembrane protein that mediates cell to cell adhesion and immune reactions by binding to the integrin lymphocyte function agent (LFA)-1 and macrophage-1 antigen [16,17,18]. A minority of RV-A strains use Low-Density Lipoprotein Receptor (LDLR) [19], and all RV-C strains characterised so far use the Cadherin Related Family Member 3 (CDHR3), as receptor [17,20,21]. (2) Uptake of the virion into the cell goes through different endocytic routes, such as macropinocytosis or clathrin-dependent or -independent endocytosis, dependent on receptor usage. (3) The virus undergoes a conformational change within the endosome, and the acidic environment (preferred method for minor-group RV’s), to form hydrophobic subviral particles [18,22,23,24]. Though the exact mechanism of release is unknown, it is suggested that the viral proteins rupture the endosome to release the genome into the cytoplasm [13,14]. (4) Translation. Once the genome and VP4 enter the cytosol, the host cell ribosomes translate the (+)ss-RNA into a polyprotein [18,25,26]. The genome is translated into a single polyprotein (P0) that is co-translationally cleaved in *cis* and *trans* by viral proteases (2A protease, 2Apro and 3C protease, 3Cpro) into 11 proteins. The initial cleavage mediated by 2Apro results in the release of P1 from the polyprotein. P1 is cleaved to form the viral capsid proteins 3Cpro is probably responsible for all the other polyprotein cleavages; releasing P2 and P3, and subsequently cleaving them to form the non-structural proteins 2Apro, 2B, 2C, 3A, 3B (VPg), 3Cpro, 3D (RNA-dependent RNA polymerase) [27]. (5) RNA replication takes place within virus-induced membranous replication organelles where 3Dpol synthesises (−)ssRNA, resulting in the formation of dsRNA, an important pathogen associated molecular pattern (PAMP). The full length (−)ssRNA then functions as the template for synthesis of new (+)ssRNA genomes that either enter another round of replication or get packaged into virions [13,14,15]. Finally, (6) assembly and release of new infectious virions. Recent studies have shown an important role for myristoylation of the capsid protein VP0 in capsid assembly [15,18,22]. The RV polyprotein is co-translationally modified at the N-terminus by myristoylation. Release of myristoylated P1 by 2Apro is followed by cleavage into VP0, VP3, VP1 by 3Cpro and formation of a protomer [28]. Protomers assemble into pentamers, that form an icosahedral capsid, enclosing the RV genome [28]. Exactly how the process ensures specificity (RV (+)ssRNA) and number of the RNA moieties being encapsidated is unclear, but VP4 domain may have a role. (7) The final step in virion maturation is the cleavage of VP0 into VP4 and VP2, by an as yet an unknown mechanism. The release of infectious virions from the cell probably varies between strains, as various pathways have been shown to be required, including cell lysis and release in membrane encased structures [13,14]. 

## 4. Cell Death Pathways

Regulated cell death is a conserved mechanism to maintain homeostasis in physiological and pathological settings. Complex intra cellular signaling pathways underpin the observed morphological changes and are triggered in response to physiological signals or injury and infection. Cell death models are classified according to their morphological appearance, enzymatic criteria, or immunological characteristics [29]. The major pathways with known involvement in the lifecycle of viruses are described below. 

### 4.1. Apoptosis

Apoptosis is characterized by cell shrinkage, nuclear condensation and plasma membrane blebbing with little or no inflammation. A characteristic feature of apoptosis is pyknosis (irreversible condensation of chromatin in the nucleus of a cell) followed by extensive membrane blebbing and then karyorrhexis ultimately resulting in the separation of cell fragments [29]. Initiated via extrinsic or intrinsic pathways, apoptosis does not involve inflammation as the cell contents are contained within the membrane bound apoptotic bodies that are immediately phagocytosed.

The extrinsic pathway (Figure 2) involves transmembrane receptor-mediated interactions. This pathway involves death receptors, proteins that contain an extracellular cysteine-rich domain and a cytoplasmic death domain. Activation of the death domain initiates a signal from the cell surface to the intracellular signaling pathways to induce apoptosis. The best characterised extrinsic pathways are the Fas Ligand/Fas Receptor (FasL/FasR) and Tumour Necrosis Factor-alpha/Tumour Necrosis Factor Receptor (TNF-*α*/TNFR1) pathways [30,31,32,33,34,35]. 

Once ligand-binding occurs (Figure 2A), cytoplasmic adaptor proteins are recruited and bind to respective death domains of the receptor proteins. Fas-associated protein with death domain (FADD), is recruited to the FasL/FasR cytoplasm domain. Bound FADD associates with pro-caspase 8, which is activated by the action of death-inducing signalling complex (DISC). Tumour necrosis factor receptor type 1-associated death domain protein (TRADD), FADD and receptor-interacting protein (RIP) are recruited to TNF-*α*/TNFR1 complex. Bound FADD associates with pro-caspase 8 leading to the formation of DISC, resulting in the activation of caspase 8. Activation of caspase-8 initiates the execution phase of apoptosis [5,30,36,37].

The intrinsic pathway of apoptosis, also known as the mitochondrial pathway, involves a variety of non-receptor-mediated stimuli that generate mitochondrial-initiated intracellular signals [5]. Positive or negative signals such as absence of certain hormones, cytokines or growth factors, presence of hypoxia, toxins, reactive oxygen species or radiation can initiate the intrinsic pathway. These signals (Figure 2B) alter the inner mitochondrial membrane, leading to opening of the mitochondrial permeability transition (MPT) pores, loss of transmembrane potential and release of two groups of pro-apoptotic proteins from the intermembrane space into the cytosol [5,36]. 

The first group consists of proteins (cytochrome *c*, Smac/DIABLO, and the serine protease HtrA2/Omi) and activates the caspase-dependent mitochondrial pathway [30,38,39,40]. Cytochrome *c* binds and activates Apaf-1 as well as procaspase-9, forming an “apoptosome” in the cytoplasm [41,42]. This complex then cleaves and activates the executioner caspases-3/6/7, resulting in apoptosis. The second group of pro-apoptotic proteins, namely AIF, endonuclease G and CAD, are released from mitochondria in the later stages of apoptosis. AIF and endonuclease G are caspase-independent proteins that translocate to the nucleus. They induce condensation of chromatin followed by DNA fragmentation [43,44]. CAD is subsequently released from the mitochondria and translocates to the nucleus where, after cleavage by caspase-3, it completes the condensation and fragmentation process [45]. The cascade of events results in destruction of the nuclear proteins and cytoskeleton, crosslinking of proteins and the formation of apoptotic bodies [30,32].

#### Rhinovirus Can Modulate Apoptosis

Infection with RV-B14 in HeLa and 16HBE14o− bronchial epithelium cells exhibited typical apoptotic morphological alterations similar to those induced by puromycin [36]. The apoptotic morphological alterations, such as cell contraction and nuclear condensation, coincided with high-molecular-weight DNA fragmentation, cytochrome c translocation from the mitochondria to the cytoplasm, activation of caspase-9 and caspase-3 and poly(ADP–ribose) polymerase cleavage. Apoptosis did not affect RV14 replication per se, but it facilitated the release of newly formed virions from cells [36]. 

The RV 3Cpro targets several cellular proteins, probably to support virus replication and limit the innate antiviral response, including apoptosis. RV 2Apro, 3Cpro, and 3CDpro have variously been shown to cleave proapoptotic adaptor proteins and nucleoporins resulting in downregulation of apoptosis and disruption of nucleocytoplasmic trafficking pathways involved in apoptosis, respectively. Nuclear transport of caspases is key to the nuclear consequences of apoptosis and it is possible that the disruption of nuclear transport in RV infected cells [48,49] may serve to inhibit the progression of apoptosis; however, this remains to be investigated [50].

RV 3Cpro cleaves RIP1, a central player in pro-inflammatory, pro-survival and cell death pathways. RV-A16 infection in HeLa cells results in activation of caspase 8 which cleaves RIP1, committing the cell to apoptosis. However, subsequent cleavage of RIP1 by 3Cpro halts the progression of apoptosis. Significantly, 3Cpro mediated cleavage was observed in cells infected with representative RV-A, -B, major and minor strains (Sarah N. Croft, personal communication), suggesting that it may be conserved across most RV strains. Intriguingly, markers of a necrotic cell death pathway were observed late in infection, along with membrane rupture [51]. Sudden release of calcium ions from their storage location in the Endoplasmic Reticulum (ER) can induce intrinsic apoptosis. The RV 2B protein inserts into the intracellular membranes and disrupts calcium stores, probably via leakage of calcium ions into the cytoplasm, reducing the calcium ions available to translocate to the mitochondria, resulting in apoptosis inhibition [52,53,54].

### 4.2. Autophagy

Autophagy is the term given to any pathway in which cytoplasmic material is transported to the lysosome [55,56]. It is a lysosomal-dependent process which is based on the degradation of the mitochondrial and other intracellular structures. Autophagy is used to maintain cell homeostasis under stressful conditions; this is achieved through the removal of misfolded protein, damaged organelles or intracellular pathogens, e.g., viruses [55].

There are three types of autophagy (Figure 3), micro-autophagy, chaperone-mediated autophagy and macro-autophagy. Micro-autophagy involves cytosolic components being directly taken up by the lysosome itself through invagination of the lysosomal membrane, and unlike macro-autophagy, engulfment in micro-autophagy is by both non-selective and selective mechanisms [57]. Chaperone-mediated autophagy involves proteins being translocated to the lysosomal membrane by chaperone complexes [57]. Chaperone proteins (i.e., heat shock cognate 70(HSC70)) and co-chaperones recognize proteins that contain KFERQ-like pentapeptides. Once the chaperones and proteins are bound, the whole complex is translocated into the lysosomal lumen and where it binds with Lysosomal-associated protein 2A (LAMP-2A) [55,56,57]. LAMP-2A, a transmembrane protein, acts as the receptor in which proteins bind for unfolding and degradation [55].

Macro-autophagy is the best characterized among the known types of autophagy (henceforth will be referred to as autophagy) [58,59]. This pathway involves the transfer of cytosolic material into lysosomes using double membrane-bound vesicles which fuse to the lysosomes [57]. 

#### 4.2.1. The Pathway and Mechanisms of Autophagy

The primary mechanism involved in autophagy can be divided into four stages: (i) induction and nucleation, (ii) elongation, (iii) cargo recruitment and (iv) fusion with the lysosome and breakdown. (i) Induction and nucleation: Autophagy is initiated when Unc-51 like kinase-1 (ULK1) complex interacts with the mechanistic target of rapamycin complex 1 (mTORC1), and AMP-activated protein kinase (AMPK) [62]. The ULK1 complex phosphorylates a class III phosphoinositide 3-kinase (P13K) complex, comprised of autophagy-related 14L (ATG14L), Beclin 1, vacuolar protein sorting 34 (VSP34), VSP15, VPS15, and autophagy and beclin 1 regulator 1 (AMBRA-1) [62,63]. AMBRA1 is phosphorylated by ULK1 after which it separates from the complex and gets translocated to the ER where it binds to Beclin-1. During the formation of the phagophore VPS34, generates PI3P, which is vital for phagophore growth [64]. The activity of VPS34- Beclin-1 complex is regulated in a Beclin-1 dependent manner by proteins like Barkor and UV resistance-associated gene (UVRAG) [64]. (ii) Elongation: Two protein conjugation systems are involved during elongation and formation of the autophagophore. One system requires the formation of the Atg12–Atg5-Atg16 complex, this system uses Atg7, Atg10 and Atg16L1. The complex is anchored onto phosphoinositol 3-phosphate on emerging autophagosomal membranes, the ATG5-ATG12-ATG16L1 complex adds to the curvature of the phagophore and is essential for the microtubule-associated protein 1A/1B-light chain 3 (LC3) lipidation [55]. The other system is the LC3-phosphatidylethanolamine (LC3-PE) system that includes ATG7 and ATG3 conjugate, LC3 (ATG8), and Atg4B [64]. ATG4 cleaves LC3 which results in the formation of LC3-1, which is then conjugated to the lipid phosphatidylethanolamine (PE) on the surface of the emerging autophagosome by ATG3 and ATG7 [62]. The final stage is the formation of LC3B-I-PE conjugate. The LC3B-I-PE conjugate elongates and seals the phagophore. LC3B-II remains associated with the autophagosomal membrane until the fusion with lysosome takes place [64]. Both systems result in a double-membraned autophagosome. (iii) Cargo recruitment: During the formation of the autophagosome, cargo is collected for degradation. Protein p60 binds to any ubiquitinated proteins and makes them a target for degradation [64]. p60 and its bound protein bind to LC3II present on the inner and outer surface of the autophagosomal membrane where it interacts with the constitutively expressed adaptor molecule P62/SQSTM1 [64]. P62/SQSTM1 and NBR1 are cargo receptors that contain a ubiquitin-binding domain [62,64]. As the autophagosome is formed, the target proteins and organelles become engulfed in the newly formed autophagosome [63]. Once the membrane closes around its cargo, LC3-II is cleaved from the outer membrane of this structure. (iv) Fusion and cargo breakdown: The final step in the autophagy pathway is the least understood [59]. Fully formed and cargo-bound autophagosomes are transported on microtubules to the perinuclear region where the outer membrane fuses with the lysosome to form an autolysosome [59,62]. The full fusion of autophagosomes occurs when lysosomal hydrolases degrade the inner membrane and expose the contents to the lysosome lumen [55], UVRAG, which associates with the PtdIns3K complex, can activate the GTPase RAB7, that promotes fusion with the lysosome [59]. In some instances, the autophagosome may fuse with an endosome and form an amphisome involving VTIlB protein, prior to fusing with the lysosome [59]. Rab7 effector protein, PLEKHM1, regulates the fusion of autophagosome and lysosome through HOPS complex and LC3/GABARAP protein [64].

#### 4.2.2. The Interactions between Autophagy and Rhinovirus Infection

The interactions between RV and autophagy are relatively understudied compared to other cell death pathways. However, it has gained attention in recent years. RV infection has been shown to induce the formation of autophagosome-like structures [65]; however, not all RV strains use autophagy in the same way or at all.

A study has suggested that RV-A2, a minor group RV, induces and uses autophagy for more efficient replication, although it is still unclear as to what autophagy machinery is involved [65]. RV-A2 infected HeLa-cells showed autophagosome formation indicated both by punctate green fluorescent protein (GFP)-LC3 signal colocalized with LAMP1 staining, and an increase in LC3-II, during the later stages of infection (4 and 6 h post infection). These two points of data confirm that RV-2 infection stimulates autophagic induction [65]. RV-A2 use of autophagy pathways was further confirmed by the demonstration that chemical inhibitors of autophagy or deletion of critical autophagy genes inhibited RV-A2 replication [65]. In contrast, an earlier study [66] found that RV-A2 did not induce the synthesis of LAMP-2- and LC3-positive compartments and modification of autophagy does not result in increased viral synthesis [66]. Klein and Jackson attribute the discrepancy in the two studies to improved understanding of autophagy in the time between the two studies as well as the types of cells and autophagosome formation markers used. However, these studies need to be further validated for different RV strains.

Two studies by Wu et al. [67,68] have shown that RV-A16 utilizes autophagy during its replication in the context of interleukin-1 receptor associated kinase M (IRAK-M). LC3 II/LC3 I protein, an indicator of autophagosome formation, was increased during RV-A16 infection in a cell line over expressing IRAK-M. Using beclin-1 to inhibit autophagy, the authors observed a marked decrease of RV-A16 RNA levels and viral particles, suggesting that autophagy may facilitate RV-A16 replication in cells in which IRAK-M is over-expressed [67]. Additionally, trehalose-induced autophagy directly inhibited Interferon lambda-1 (IFN-λ1) expression and promoted RV-A16 infection in normal human primary airway epithelial cells. This effect was reversed when ATG5 was knocked out. Trehalose is a natural glucose disaccharide which functions in the prevention of LPS-mediated inflammatory response [69] and induces autophagy in various cells by promoting the recruitment of LC3 II into the forming autophagosome membrane in an ATG5-ATG12-dependent manner [70]. Interestingly, RV-A16 infection was shown to result in reduced levels of p62/SQSTM1 late in infection, probably due to its cleavage by 3Cpro [51]; effects on autophagy were not investigated in the study.

RV-B14 infection may also induce autophagy as shown by co-localization of LAMP1 with punctate GFP-LC3 in MCF-7 cells. Further, monodansylcadaverine, a fluorophore retained in autophagosomes under mild cell fixation conditions, showed a punctate pattern after infection with RV-B14 and suggesting the formation of autophagosome-like vesicles [65].

In contrast, RV-A1a does not induce or use autophagic signaling or autophagosomes [65]. In the same study in which RV-A2 was found to induce LC3 modification, simulating autophagy, RV-A1a did not induce LC3 modification or simulate autophagy [65]. No other RV strains have been investigated for their exploitation and/or modulation of autophagy.

### 4.3. Necrosis

Necrosis follows an energy-independent mode of death that occurs in an unprogrammed state in cells induced by a number of external triggers such as infection, trauma, or excessive stress. It is a pathological event characterised by cytoplasmic swelling, irreversible membrane damage and organelle breakdown, with characteristic cellular content leakage into the extracellular environment. Leaked cellular content triggers an increase in the secretion of pro-inflammatory cytokines from activated macrophages [71]. 

Necrosis triggered by DNA damage leads to the activation of poly (ADP-ribose) polymerase 1 (PARP-1) to catalyze the hydrolysis of NAD+ into nicotinamide and poly-ADP ribose, causing NAD depletion. This results in cellular energy failure and a caspase- independent death. N-methyl-N-nitro-N-nitrosoguanidine (MNNG)- induced necrosis is dependent on RIP-1 and TNFR associated factor (TRAF)2, which function downstream of PARP-1 and lead to c-Jun N-terminal kinase (JNK) activation. JNK affects mitochondrial membrane integrity which allows for the release of proteins into the mitochondrial intermembrane space. The specific way JNK induces this is unclear however, either modification of Bcl-2 family or caspase-independent JNK-mediated processing of bid may be at play [72].

#### Modulation of Necrosis by Rhinovirus

Limited studies have looked into the modulation of necrosis by RV’s. Lötzerich et al., 2018 have shown 3Cpro is able to suppress apoptosis and trigger necrosis [51]. Montgomery et al., 2020 have shown a direct correlation between RV infection, increased expression of IL-1 and increased necrotic events [73]. RV was shown to induce necrosis in airway epithelial cells collected from children with and without cystic fibrosis accompanied by higher levels of inflammatory mediators IL-1α, IL-1β and IL-8 [73], suggesting necrosis as a mechanism by which RVs induce mucin accumulation and inflammation in early lung disease. Whether necrosis has a role in virus assembly or release is unclear.

### 4.4. Necroptosis

Necroptosis is a programmed variant of necrosis triggered by interactions of death ligands and death receptors in the context of caspase inhibition. Necroptosis and necrosis share morphological features such as early destruction of membrane integrity, cell and intracellular organelle swelling, oxidative bursts and cell content spillage. Necroptosis leads to reduced damage to the cells by regulation of a series of signals [74]. Necroptosis, unlike apoptosis, causes local inflammation, infiltration and activation of inflammatory cells. This inflammatory process mediates the release of damage-associated molecular patterns (DAMPs), which results in the recruitment of pro-inflammatory cell types to the sites of infection. The necroptotic pathway has three key players; RIP1, RIP3 and mixed linkage kinase domain-like protein (MLKL) [75].

#### 4.4.1. Necroptosis Pathway

Necroptosis is induced by a class of death receptors that include TNFR1, TNFR2 and Fas. Upon binding with their respective agonists, the cell can be directed down a cell death pathway. Most research into the necroptotic pathway has been done on the TNFa/TNFR1 induced pathway (Figure 4). TNFa bound to the extracellular domain of TNFR1 creates allosteric changes in the cytoplasmic region of TNFR1, triggering downstream signaling by forming complex I with TRADD, FADD, RIP1, and E3 ubiquitination ligases, TRAF2/5 and cellular inhibitor of apoptosis protein-1 (cIAP1/2) [68].

RIP1, a serine/threonine kinase, plays a role in both necroptosis and apoptosis (in addition to necrosis), and is initially recruited to complex I by TNFR1 and is polyubiquitinated by TRAF2/5, cIAP1/2 on lysine at position 63; this drives the cell down a pro-survival pathway. This pathway progresses via Inhibitor of Kappa B Kinase (IKK) and NF-kappa B Essential Modulator (NEMO) recruitment and activation of NF-kB pathway. However, when RIP1 is deubiquitinated, it inhibits the NF-kB pathway, which directs the cell towards death [68,76].

Deubiquitinated RIP1 is transferred from complex I to the cytoplasm and prompts the recruitment of complex II. TRADD is also released from complex I at this time. Complex II, or DISC, comprises TRADD, FADD, RIP1 and caspase 8. Caspase 8 is inhibited in complex II, allowing RIP1 to bind to RIP3. Bound RIP3 recruits and phosphorylates MLKL at serine 358 and threonine 357 resulting in the formation of the necrosome [77]. Activated MLKL then binds to phosphatidylinositol phosphates (PIPs), Cardiolipin (CL) and phosphatidylglycerol (PG), allowing the necrosome to move through the cell and target the membrane where MLKL disrupts the membrane leading to permeabilization, swelling and rupturing [78]. The final or propagation stage of necroptosis involves an inflammatory wave of DAMP release [76]. DAMP production is a crucial contributor to chronic and acute inflammation, specifically induction of cytokine and other chemo-attractants, for the recruitment of primary immune cells to the necroptotic site [76]. The steps between complex II and the formation of the necrosome as well as the downstream signaling in the necroptotic pathway are still under-researched.

#### 4.4.2. Potential Exploitation of Necroptosis by Rhinovirus

The pathway used by RV to exit from a cell after replication is still unknown however there are several lines of direct and indirect evidence to suggest that necroptosis may be used during RV infection. Recent studies [36,37] have identified that RV infection inhibits caspase-dependent cell death pathways like apoptosis through the cleavage of RIP1 by 3Cpro, coinciding with impaired recruitment of active caspase 8 [37,51]. Inhibition of caspase 8 activity would be expected to commit the cell to necroptosis. Whether the 3Cpro cleavage product of RIP1 is able to interact with RIP3, an essential step in necroptosis, is unknown. IL-33, a key necroptotic inflammatory factor, has been shown to be released during RV infection. RV infection triggers the production of several cytokines and chemokines (including IL-1, IL-6, IL-8, GM-CSF, eotaxins, and regulated upon activation normal T-cell expressed and secreted (RANTES), TNF-α, IFN-γ, and macrophage inflammatory protein (MIP)–1a [83,84,85]). Among these, TNF-α and IL-1 are known to induce the necroptotic pathway [86]. This surge in expression of cytokines and chemokines leads us to speculate that RV-induced inflammation may trigger necroptosis within the cell. However, further studies are required to address this link in more detail. Elevated cytosolic calcium is observed during RV infection, probably due to the actions of 2B protein (see above). Accumulation of cytosolic calcium ions led to inhibition or deficiency of caspase 8 in neuroblastoma cells, along with Ca2+/calmodulin-dependent protein kinase II (CaMK II) phospholrylation and necroptosis [54]. The RV viroporin (2B) disrupts the calcium ion homeostasis within the host cell by discretely affecting calcium release from the ER [50]. This suggests that necroptosis may be a potential viral release mechanism for RV-A2 through the accumulation of cytosolic Ca2+ within infected cells [87]. Finally, a study has shown that chemical inhibition of membrane channels characteristic of necroptosis inhibits RV-A2 release [88]. Together, the evidence provides a strong incentive to investigate the possible role of necroptosis in RV release. Recently it has been seen that a deficiency in IFN-β, a multipurpose cytokine, during asthma exasperations promotes necroptosis. Primary cells from asthmatics show a deficient ability to produce IFN-β during RV infection, and RV induced asthma exacerbations have been linked to increased levels of LDH [89]. Results from experimentation on mice deficient in IFN-β suggest that necroptosis may contribute to high LDH levels due to the observed increase in pMLKL [90]. Data from this paper implies that RV infection may result in a more harmful cell death response in asthmatics with reduced IFN expression. However, further studies should be conducted into the role of IFN-β as a regulator of necroptosis during RV infection.

### 4.5. Parthanatos, Stress and Rhinovirus

Parthanatos is a relatively new addition to cell death pathway characterized in recent years. The pathway is regulated by poly(ADP-ribose) polymerase 1 (PARP1), an enzyme that catalyzes DNA base excision repair [91]. PARP1 is activated in response to severe or prolonged DNA damage, oxidative stress, hypoxia, hypoglycemia or inflammation. This triggers a series of cytotoxic effects such as NAD+ and ATP depletion and accumulation of poly(ADP-ribose) polymers (PAR) and poly(ADP-ribosyl)ated proteins (PARP) at mitochondria (Figure 5) [92,93]. RV-1B infection of human primary bronchial epithelial cells (pBECs) exposed to cigarette smoke or other oxidative stressors reduced mitochondrial respiration, increased proton leak and subsequently increased pro-inflammatory cytokines [94]. The oxidative stress caused by the external stimuli, including RV infection, lead to increased expression of cytochrome C [94]. Exposure of RV-A16 infected human bronchial epithelial cells and the BEAS-2B cells to cigarette smoke increased the expression of CXCL8 (linked to mRNA stabilization) compared to either stimulus by themselves. Cigarette smoke was shown to inhibit RV-16-induced expression of CXCL-10 vis transcriptional regulation [94,95]. This suggests parthanatos could be one of the mechanisms by which RV infections, aggravated by external stimuli such as cigarette smoke, can induce exacerbations in asthma and chronic obstructive pulmonary disease (COPD) [94,95]. This could potentially explain increased susceptibility of smokers, asthma and COPD patients to respiratory infections and contribute to exacerbations.

## 5. Conclusions 

RVs have significant impact on society being the primary agent for upper respiratory tract infections globally. The indications range from mild (common cold) limited to the upper respiratory tract, to chronic or severe illnesses (such as, asthma exacerbations, bronchiolitis, and otitis media) [9]. Vaccine design against rhinoviruses is unlikely due to its high mutation rates and over 100 serotypes while existing non-specific antiviral treatments are generally ineffective. Characterization of how the virus subverts cell signalling pathways, especially cell death, could potentially drive targeted drug development [97]. Most of the work to date has focussed on the interplay between RVs and apoptosis, with some studies examining necrosis and autophagy. The review of existing direct and indirect evidence presented here clearly shows that RVs have a very complex relationship with several cell death pathways, opening up several possible lines of future research investigation.

That RVs manipulate cell death pathways to favour viral replication and propagation, is clear [32,98]. This manipulation may correlate with the severity of disease and exacerbations of pre-existing lung conditions [10,20,97]. Our understanding of RV interaction with and modulation of cell death pathways has evolved along with our knowledge of these complex pathways. Accumulating literature shows that RVs inhibit apoptosis early in infection via cleavage of RIP1 and by disrupting nucleocytoplasmic trafficking; although the latter has not been directly linked with apoptosis inhibition. Recent studies implicate necroptosis as a potential mechanism by which RVs exit the host cell, accompanied by release of proinflammatory chemokines and cytokines. RIP1, which is cleaved in RV infection, is a master regulator, positioned to commit the cell to prosurvival, apoptotic or necroptotic pathway. RV infection induces caspase 8 dependent apoptotic pathway, which is quickly inhibited by cleavage of RIP1, possibly committing the cell to necroptosis. This supposition is supported by the observation of necrotic markers in the absence of late-stage necrosis in RV infected cells. RVs appear to modulate autophagy in a strain, maybe species, specific manner. The limited literature raises the question—does the autophagy pathway facilitate RV release, albeit inside a membrane vesicle? Chronic damage to DNA caused by extensive oxidative stress is shown to induce parthanatos during RV infections. Parthanatos is one of the mechanisms by which RV infections may induce exacerbations in asthma and COPD. Intriguingly, cell death related targets of 3Cpro identified so far (e.g., p62/SQSTM1) are involved in multiple pathways, possibly suggesting a fine modulation of several pathways by RVs that may vary through the lifecycle to enable optimal virus replication, assembly and release. A limitation for our review was the lack of studies on modulation of cell death pathways by RVs belonging to genotype C. However, review of available literature on RV-A and RV-B strains strongly suggests the role of necroptosis and autophagy in RV release. 

## Figures and Tables

**Figure 1 viruses-13-00629-f001:**
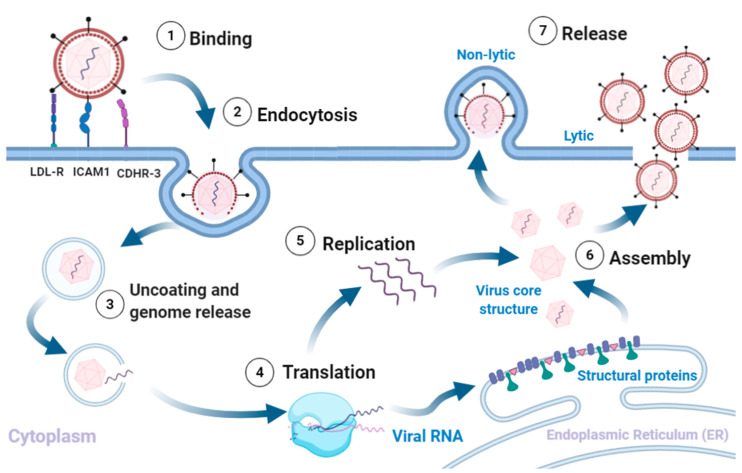
Rhinovirus lifecycle. (1) Binding of the virus to receptors on the plasma membrane initiates (2) endocytosis. The intake of the virion is followed by (3) uncoating of the capsid and release of the (+)ss-RNA into the host cytoplasm, which is (4) translated and processed into structural and non-structural proteins. The viral RNA dependent RNA polymerase converts the viral genome into (−)ss-RNA, that is (5) replicated into new (+)ss-RNA genomes. The new genomes become the template for translation into new viral proteins and replication into new genomes. The final stages involve (6) assembly of the structural proteins and RNA genome into capsids and (7) release of new infectious virions via lysis or non-lytic mechanisms. Adapted from [5,13,16,28]. Created with BioRender.com, accessed on 12 December 2020.

**Figure 2 viruses-13-00629-f002:**
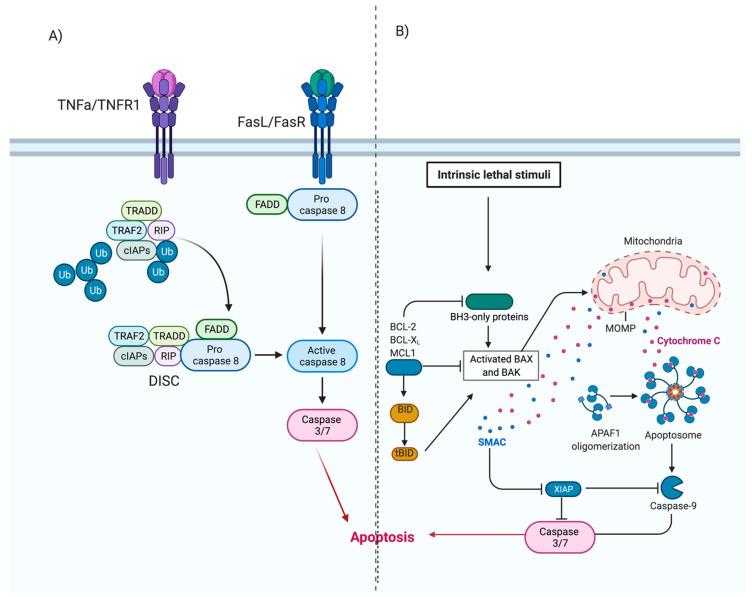
Extrinsic and Intrinsic apoptosis. (**A**) Extrinsic apoptosis. Ligand binding results in activation of cytoplasmic adapter proteins and recruitment of death domains on FasL/FasR or TNF-α/TNFR1. This activates FADD, which can directly activate pro-caspase-8 or can form a complex with TRADD and RIP, bind to TNF-α/TNFR1 and then induce auto-catalytic activation of procaspase-8. Activated caspase 8 initiates the execution phase of apoptosis. (**B**) Intrinsic apoptosis. Initiated by non-receptor-mediated stimuli that generate mitochondrial-initiated intracellular signals. These signals initiate a cascade of events resulting in the formation of an apoptosome. The apoptosome activates the executioner caspases-3/6/7 and initiates apoptosis. Adapted from [5,46,47]. Created with BioRender.com, accessed on 12 December 2020.

**Figure 3 viruses-13-00629-f003:**
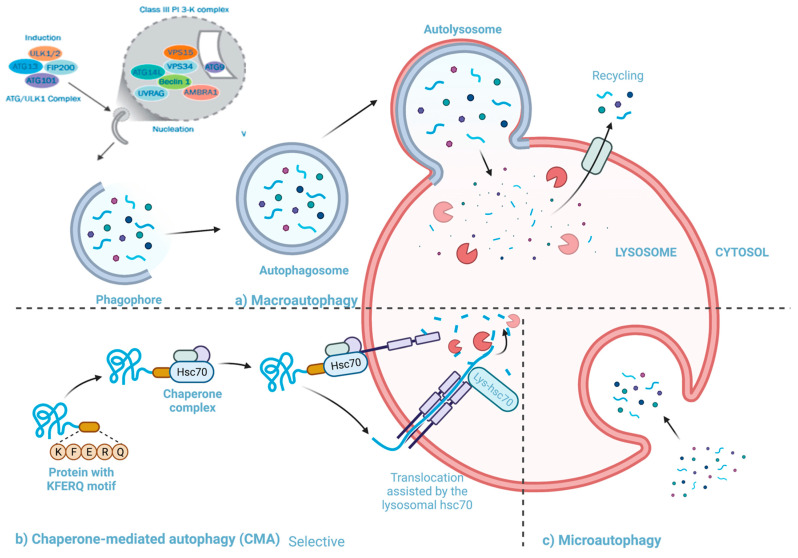
Three autophagy pathways (**a**) Macroautophagy: cytosolic material is transported to the lysosome using double membrane-bound vesicles which engulf the material and then fuse to the lysosome (**b**) Chaperone mediated autophagy: proteins bound to HSC70 being translocated to the lysosomal membrane to be unfolded and degraded. (**c**) Microautophagy: cytosolic components being directly taken up by the lysosome. Adapted from [60,61]. Created with BioRender.com, accessed on 25 January 2020.

**Figure 4 viruses-13-00629-f004:**
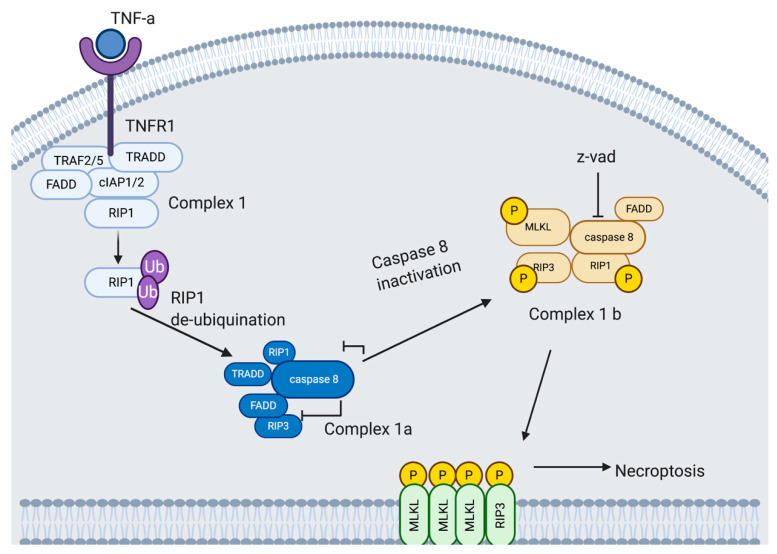
Necroptopic signaling pathway. TNFa is bound to the extracellular domain of the TNFR1. TNFR1 triggers downstream signaling by forming complex I with TRADD, FADD, RIP1, and several E3 ubiquitination ligases, TRAF2/5 and cIAP1/2. Deubiquitinated RIP1 is transferred from complex I to the cytoplasm and prompts the recruitment of complex Ia. Complex Ia is comprised of TRADD, FADD, RIP1 and caspase 8. Caspase 8 is inhibited in complex Ib, allowing for RIP1 to bind to RIP3. Bound RIP3 recruits and phosphorylates MLKL. Activated MLKL disrupts the membrane leading to permeabilization, swelling and rupturing. Adapted from [51,79,80,81,82]. Created with BioRender.com, accessed on 15 January 2021.

**Figure 5 viruses-13-00629-f005:**
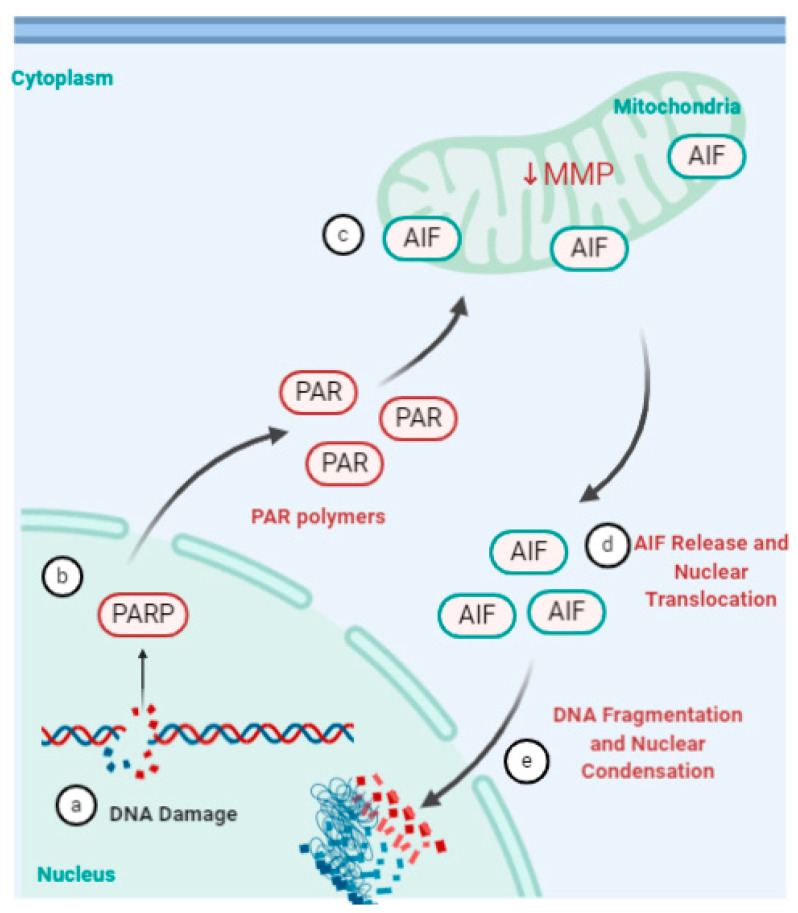
Parthanatos pathway. (a) Severe or prolonged DNA damage, oxidative stress, hypoxia, hypoglycemia or inflammation (b) triggers the activation of the enzyme poly(ADP-ribose) polymerase (PARP). (c) This triggers a series of cytotoxic effects such as NAD+ and ATP depletion and accumulation of poly(ADP-ribosyl)ated proteins (PAR) at the mitochondria. (d) PAR bind to apoptosis-inducing factor (AIF), decrease in mitochondrial membrane potential (MMP) and prompts release of AIF into the cytoplasm. (e) AIF then translocates to the nucleus and mediates large-scale DNA fragmentation and chromatin condensation [91,92,93,96]. Created using BioRender.com, accessed on 31 January 2021.

## Data Availability

Not applicable.

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
