# Peer review of "Rhinovirus and Cell Death"

_viruses, 2021, doi:10.3390/v13040629_

Round 1
Reviewer 1 Report
The review is well organized
Author Response
We thank the reviewer for the positive comment on our manuscript.
Reviewer 2 Report
This is an interesting review and it is very comprehensive. My only suggestion for improvement is to add a glossary of specific terms, e.g. pyknosis, karyorrhexis etc
Author Response
We have noted the reviewer’s comment and have included the definition of the specific terms within the text, e.g. section 3.1 - A characteristic feature of apoptosis is pyknosis (irreversible condensation of chroma-tin in the nucleus of a cell)…
Reviewer 3 Report
The manuscript by Kerr et all reviews the interplay between rhinoviruses and cell death. After reviewing the molecular biology and replication of RV, the mechanisms underlying five different cell death pathways are reviewed: apoptosis, autophagy, necrosis, necroptosis and parthanatos. Following a description of each cell death pathway the literature examining the interaction of RV with these pathways is reviewed and discussed in relation potential roles in the viral life cycle and/or pathogenesis. These are based on a literature search going back 21 years. The review is accompanied by high quality figures that aid in understanding the various death pathways being discussed. Overall, this is an excellent review that will be useful for those focused on how RV and other viruses modulate cell death pathways. By addressing a few issues the manuscript could be improved further. First, include a better description of how the literature search was conducted- what keywords were used? Second, avoid generalizing study results as reflective of all rhinoviruses. For the most part the authors do an excellent job of this, but there are a few examples where the species/strain should be indicated (e.g., line 400- indicate RV2). Saying that VPg functions similar to the 5’ cap (line 56) isn’t quite correct and could give readers the mistaken impression that it has a role in recruiting translation preinitiation complex. When describing the processing of the viral polyprotein, include mention that in addition to the final products, precursors such as 3CD, etc. may have distinct roles the viral life cycle. The References should be reviewed carefully for accuracy as there were a few apparent errors. For example, lines 40 and 63 refer to references 5 and 6 to support an overview of RV pathogenesis and receptor usage. Line 132 cites reference 33 for review of the extrinsic pathway, Reference 48 and 83 are duplicated.
Minor comments:
Figure 2- The two panels use different icons for caspase 3/7, which could be confusing. Modiify to use the same icon in each panel.
Line 395-399- clarify that caspase 8 deficiency was not observed in rhinovirus-infected cells in cited reference. Further explain the suggested possible mechanism as it was not clear to me.
Figure 5- indicate labels present in figure (a-e) in figure legend.
Line 468-9- change from ‘can induce exacerbations’ to ‘may induce exacerbations
Author Response
The manuscript by Kerr et all reviews the interplay between rhinoviruses and cell death. After reviewing the molecular biology and replication of RV, the mechanisms underlying five different cell death pathways are reviewed: apoptosis, autophagy, necrosis, necroptosis and parthanatos. Following a description of each cell death pathway the literature examining the interaction of RV with these pathways is reviewed and discussed in relation potential roles in the viral life cycle and/or pathogenesis. These are based on a literature search going back 21 years. The review is accompanied by high quality figures that aid in understanding the various death pathways being discussed. Overall, this is an excellent review that will be useful for those focused on how RV and other viruses modulate cell death pathways.
We thank the reviewer for the positive comment on our manuscript.
By addressing a few issues the manuscript could be improved further. First, include a better description of how the literature search was conducted- what keywords were used?
We have now expanded the description of the literature search methodology – see section 2. Search Strategy in the revised manuscript.
Second, avoid generalizing study results as reflective of all rhinoviruses. For the most part the authors do an excellent job of this, but there are a few examples where the species/strain should be indicated (e.g., line 400- indicate RV2).
The Reviewer’s point is well taken. We have now changed all study descriptions to refer to the specific strains used in the individual studies, wherever the information was available.
Saying that VPg functions similar to the 5’ cap (line 56) isn’t quite correct and could give readers the mistaken impression that it has a role in recruiting translation preinitiation complex.
The statement referring to VPg has now been modified accordingly.
When describing the processing of the viral polyprotein, include mention that in addition to the final products, precursors such as 3CD, etc. may have distinct roles the viral life cycle.
We have decided not to include a detailed description of the various protein products formed as we felt that was not the focus of this review. The cited publications in this section contain the details of the polyprotein processing in rhinovirus infection.
The References should be reviewed carefully for accuracy as there were a few apparent errors. For example, lines 40 and 63 refer to references 5 and 6 to support an overview of RV pathogenesis and receptor usage. Line 132 cites reference 33 for review of the extrinsic pathway, Reference 48 and 83 are duplicated.
We have now carefully reviewed all the in-text citations and the bibliography and made required changes. The error in the original submission is regretted.
